# The More Fertile, the More Creative: Changes in Women’s Creative Potential across the Ovulatory Cycle

**DOI:** 10.3390/ijerph18105390

**Published:** 2021-05-18

**Authors:** Katarzyna Galasinska, Aleksandra Szymkow

**Affiliations:** 1Institute of Psychology, SWPS University of Social Sciences and Humanities, 03-815 Warsaw, Poland; aszymkow-sudziarska@swps.edu.pl; 2Center for Research on Biological Basis of Social Behavior, SWPS University of Social Sciences and Humanities, 03-815 Warsaw, Poland

**Keywords:** ovulatory cycle, creativity, signaling theory, women, fertility

## Abstract

Creative thinking is a defining human feature. It provides novel solutions and as such undoubtedly has contributed to our survival. However, according to signaling theory, creativity could also have evolved through sexual selection as a potential fitness indicator. In our study, we tested one implication of this theory. Specifically, we hypothesized that if creativity can serve as a signal of women’s fitness, then we should observe an increase in creative thinking in the fertile phase of the ovulatory cycle compared to other non-fertile phases. In our study (*N* = 751), we tested creative potential throughout the ovulatory cycle. We found a positive correlation between the probability of conception and both creative originality and flexibility. Importantly, we also tested the mediating role of arousal in the relationship between the probability of conception and creative thinking. The results of our study are discussed in terms of signaling theory, through which women advertise their fitness with their creativity.

## 1. Introduction

Ovulation is a biological process that is essential for reproductive success [1]. Its occurrence signifies the peak period of reproductive fertility and the highest probability of conception [2]. The optimal fertilization window for women appears about 12–14 h after ovulation [3]. The probability of conception occurring outside the fertile window is virtually zero [4]. Female fertility is highly dependent on age, and the decrease in fertility accelerates after the age of 35 due to reductions in the ovarian reserve and oocyte quality [5]. Human oocyte aging is also affected by various behavioral factors, such as dietary habits and lifestyle [5]. Also of importance are environmental factors such as air pollution [6] or pesticides (e.g., Mancozeb), which have been shown to interfere with female sex hormones and cause disruption of the ovarian cycle [3,7]. Thus, nowadays, lower levels of female fertility are a widespread health concern [8]. On the psychological level, according to recent evolutionary models, fertility status may affect women’s mating psychology [9,10]. For a couple of days in the middle of the menstrual cycle, women’s sensitivity to a potential mate is supposed to increase [11]. Some studies indicate that they experience both a greater arousal [12] and sexual desire, and report more frequent sexual fantasies [13,14,15]. Women show a higher openness to uncommitted sexual relationships during ovulation [16] and preferences for more traditionally masculine features in men [17]. However, this picture is not so clear-cut, as other studies do not seem to support such effects of the menstrual cycle. Thomas et al. [18] found that fertility status is unrelated to sociosexual attitudes and desires. Furthermore, contradictory findings regarding ovulatory shifts in mating preferences directing women’s attention and desire have been presented [15,19,20,21]. Nevertheless, researchers show evidence of multiple cues related to women’s fertility in reviews, see [22,23]. According to these findings, during fertility peak, when estradiol levels are highest, women’s features visibly change: their eyes become bright and their cheeks red [24]. However, some research indicates no subtle facial changes produced by estradiol [25]. Additionally, the waist-to-hip ratio decreases during the time of ovulation [26]. Importantly, these changes seem to be noticeable, as, according to some studies, ovulating women are perceived as more attractive by their partners, by other men, and even by other women [27,28,29]. Additionally, women’s body movements [30], body scent [31] and increased vocal pitch [32] are assessed by men as more attractive at the time of ovulation compared to non-fertile phases of the ovulatory cycle. During ovulation, women care for their appearance significantly more, trying to attract a potential partner with self-ornamentation [33], a more fashionable style [33], and revealing clothes [9]. Women seem to be more motivated to engage in mating behaviors, as they are more likely to attend large social gatherings where they can potentially meet new romantic partners [34]. They generally walk more [35] and eat less [36], and they are more prone to risky behavior [37]. During ovulation, women also show increased intrasexual competitiveness. For instance, they are more likely to dehumanize other women as conception risk rises [38] and they display attentional bias toward ornamental objects when primed with attractive rivals [39]. Additionally, research indicates that women are capable of detecting subtle fertility indicators in other women [40]. They may monitor other women’s fertility and adjust their own mating strategies accordingly. This appears to be indicated by the fact that detecting the fertile phase of the ovulatory cycle in other women via olfactory cues leads to the higher level of testosterone than when detecting the infertile phase [41].

One of the cognitive changes that women display during ovulation relates to creativity, namely to divergent thinking. Krug [12,42,43] found that during the ovulatory period women become more creative. However, the mechanism of this effect is unclear. In our study, we aimed to replicate previous findings and investigate the potential mediating role of arousal in the relation between the conception probability and creative potential. We argue that enhanced creativity during the fertile window of the ovulatory cycle can have adaptive functions, as according to signaling theory, creativity could have evolved through sexual selection as a potential fitness indicator [44].

### 1.1. Signaling Theory

Darwin [45] noticed that some traits, such as the elaborate colors of a peacock’s tail, might have evolved because they enhanced an individual’s ability to attract a mate. Darwin realized that evolution is driven by sexual selection through mate choice as equally as by natural selection for survival. These specific traits that successfully attracted mates were therefore selected in the process of sexual selection [44]. Based on this point of view, many of our physical and mental traits might have evolved through sexual choices [44].

The peacock’s tail is a salient example of a sexually selected trait. It attracted peahens, so it helped peacocks adapt for courtship. However, in other life domains, it may actually be detrimental. Such a large attribute definitely did not help them to escape or hide from predators. It may also have influenced the aggression of other peacocks, increasing their competitiveness. Having such a large and showy tail seems to be nutritionally costly and energetically inefficient [46]. Only highly fit peacocks could afford it, as others would not be strong enough to maintain it in demanding everyday situations. Therefore, sexual ornaments such as this serve as honest signals of the quality of one organism’s genes to potential mates [47,48,49]. Such displays have been well documented in animal studies; for review, see [50].

Darwin [45] was the first to suggest that several distinctive features of human culture, including poetry, painting and music, evolved through sexual selection in a similar way to ornamentation and birdsongs. His suggestion was developed by Miller [44] into a hypothesis indicating that human creativity can be such a signal of genetic quality. Miller [44] argues that artistic ornamentation beyond the body extends the natural sexual human adornments such as penises, beards, breasts and buttocks. He also noticed that human mate choice is focused on the mind even more than on the body, as cognitive abilities reflect genetic and environmental influences on brain development [51]. These abilities are also important cues for choosing mates among nonhuman vertebrates; for review, see [51]. For instance, satin bowerbirds with good problem-solving abilities are more likely to be chosen as partners [52]. Among lyre birds, females prefer males that mimic the sounds of other birds and add them to one’s own repertoire [53]. Nowicki, Peters and Podos [54] pointed out that large and complex repertoires of birdsongs may indicate a history of good nutrition, and females might select males with complex song repertoires for this reason. A good fitness indicator must then reliably advertise good genes that can be passed to offspring, thereby giving them attractiveness and talent. To be honest signals, traits must also ensure reproductive success [55]. Does creativity translate into reproductive success? In the light of the existing studies it does, but nearly exclusively for men. For instance, Clegg, Nettle and Miell [56] studied a large sample of British artists and found that more professional artists had more sexual partners, and this was found only for males; see also [57]. However, as the authors noticed, the non-significant finding for female artists may not mean that they are not attracting sexual partners through their creativity, but rather that they are seeking quality over quantity [56]. This is in line with studies by Griskevicius et al. [58], who found that women’s creativity increased when primed with a stimulus representing a long-term, high-quality partner but not when primed with a short-term mate. In addition, there is also evidence that creative people tend to be less likely to marry and have fewer children even when they do marry [59]. As Feist [60] argues, time spent on creative projects may be taken away from mating and child-rearing. However, even though the existing empirical evidence does not provide clear-cut conclusions here, there are reasons to predict a significant role of creative signals in mating for both men and women.

### 1.2. Courtship Displays Based on Parental Investment

Self-ornamentation and other advertising features refer to intrasexual competitiveness for access to valuable high-investing members of the opposite sex [61]. Among many species, males are usually more ornamented, larger and more competitive than females. This sexual dimorphism is dependent on the parental care provided by both sexes. According to parental investment theory [62], a greater-investing sex produces fewer offspring over the lifetime. Usually, it is a female who, as a result, tends to be choosier about her sexual partners, as her investment in such cases needs to be wiser [63]. Males, as less sex-investing, produce more offspring, so their strategy involves pursuing multiple matings. Thus, they constantly compete for mates, signaling their own value in a way that attracts females. Among these species, males often have a stronger sex drive and greater interest in sexual novelty [63]. There are also species where it is the female who is larger, more ornamented, aggressive and competitive for access to males [64]. In these species, males are choosier about their sexual partners because their investment is higher [63]. However, there are also monomorphic species that do not show these differences between the sexes. Among these species, both parents provide similar parental care. Females and males form pair bonds, and their mate choice is mutual [65]. This pattern is commonly found in birds [65] and mammals [63]. Stewart-Williams and Thomas [63] suggest that this pattern is relevant to humans as well. Human sex differences are usually modest and are stronger within one sex than between sexes and mostly concern physical traits, such as muscle mass and strength [66]. Psychological differences are much smaller [67]. Men are usually involved in raising their offspring, as the young require an enormous amount of investment, much more than the mother can provide alone [63].

As both human sexes participate in raising offspring, they both have to compete for desirable members of the other sex [63,68]. Men are similarly selective as women, especially in regard to long-term mating [68,69]. If they decide to commit, they exert greater selectivity and have elevated standards in their choice of mates: they care for the intelligence, personality traits and social reputation of a prospective mate [61]. Men do not want to take risks and commit to women low in mate value. Men’s mate preferences put women under intersexual selection and have led to the evolution of salient secondary sexual features: breasts, facial neoteny and figure [63]. Stewart-Williams and Thomas [63] considered that women may be even more sexually selected than men: they signal their attractiveness to a greater extent than men, and the competition among women takes many forms. The authors cited examples of women competing for men in writing amusing letters, playing the piano or learning foreign languages. It appears that women use creativity and intelligence as a signal of their suitability as a spouse, according to signaling theory [44]. Miller [44] emphasizes that in a mating market, both sexes select potential partners based on indicators of heritable fitness, and fitness matching based on creative courtship behavior may have been the mechanism of sexual selection which has led to human mental evolution. Additionally, there are relatively few evolved gender differences in mating strategies and preferences [70].

### 1.3. Creativity as a Signal

Creativity is essential to the development of human civilization and cultural life [71]. The ability to be creative is one of the defining human features [53]. However, the majority of psychological research on creativity does not concern its evolutionary origins. These origins are common for many species; thus, looking for creativity from an evolutionary perspective involves the identification of related behaviors in other species [53]. Creativity is a domain associated with aesthetics, so it is supposed to capture attention starting at the level of the senses [72]. In this vein, Darwin considered that the subjective sensual experience is similar among both people and non-human animals [73]. Stimulating the senses is an important means of communication for many nonhuman vertebrates, as they have no other system capable of conveying rich ideas [44]. In humans, cognitive abilities are much more advanced. Researchers exploring human creativity have emphasized that this ability provides novel and appropriate solutions [74]. Novelty requires taking different approaches relative to prevailing modes of thought or expression that are based on divergent thinking [75]. This kind of thinking allows an individual to generate many alternative ideas based on unique associations and connections [76]. It may result in diverse potential solutions to problems and is applied in both art and scientific innovations [77,78]. Taking a different point of view leads to originality, which is the major factor of creativity [79,80]. The domain involving the originality of ideas and aesthetics is where creativity separates from traditional intelligence [80], although these constructs are correlated [81]. In contrast, IQ requires convergent thinking, which leads to a correct solution, regardless of whether it is original or not [80]. Creative products, as opposed to intelligent products, do not have to be pragmatic, as they serve beauty [73,82,83,84]. Research on creativity became popular since Guilford [85] emphasized the prevalent features of the creative mental process: its fluency and flexibility results in original ideas. Divergent thinking represents a creative potential that may lead to a variety of creative realizations [80], being a domain-general creative ability [86]. As researchers have shown, creative thinking, as estimated from tests of divergent thinking, is more important in the natural environment than tests of IQ or academic tests [80].

Looking for evolutionary origins of creativity, Miller [44] proposed that creativity evolved through sexual selection and enhances an individual’s ability to attract a mate independent of whether it directly enhances survival [44]. Dutton [87] similarly claimed that sexual selection became a root of the human “art instinct”. Prum [73] emphasized an openly aesthetic character of Darwin’s theory of sexual selection as its most noteworthy feature. Feist [60] argues that natural selection has driven survival benefits such as advances in science and engineering, whereas sexual selection has driven more ornamental or aesthetic aspects of creativity, including art, music, dance, and humor. Is creativity indeed attractive? As a few authors have noted, there is no doubt that creativity is sexy [88]. Considering people’s preferences, creativity can be found among the top ten most desired traits for men and women worldwide [68], and although it is not a necessity, it is highly valued [89]. The preferences are also context-dependent. Apparently, both sexes consider the ornamental/aesthetic forms of creativity as more sexually attractive, compared to applied/technological forms [88], with females showing a stronger preference than males for ornamental/aesthetic creativity (e.g., writing music, writing poetry), and males showing a stronger preference than females for everyday/domestic forms of creativity (e.g., interior decorating, making clothes). Women value creative men in the short-term context, particularly when they are in the fertile phase of their menstrual cycle, as found by Haselton and Miller [34]. This research is the only study so far providing evidence that creativity may serve as a marker of good genetic quality [90].

### 1.4. Ovulatory Shifts and Creativity

Psychological orientation directed towards mate attraction can facilitate behavioral signaling of mental traits [13]. Roney and Simmons [91] argue that sexual hormones can be the input cues predicting mating behavior by influencing sexual motivation. Thus, during a fertile phase of the ovulatory cycle, women may display their creativity, which is what researchers have actually found, indicating that in ovulation, women are more creative than in other phases of the ovulatory cycle [12,42,43]. It has also been found that EEG dimensional complexity is higher during the fertile phase, which is an essential prerequisite for creative thinking [43]. However, there is a lack of evidence concerning the possible mechanism of this effect. Krug [12] suggested that arousal may be a possible mediator, but this possibility was not tested. Women around ovulation show stronger arousal on a physiological level [12], and arousal has been associated with increased motivation, as pleasant and unpleasant arousal reactions provide bodily information about the importance of what is experienced and refer to the activation of the sympathetic nervous system [92]. It has also been shown that estradiol, which is a hormone responsible for an ovulation peak, correlates with dopamine, an arousing reward transmitter responsible for increased motivation [93]. Thus, it seems plausible that the probability of conception is positively associated with creativity with arousal as a mediating variable.

The purpose of our study was to replicate the effect of enhanced creativity and show that creative potential is related to the probability of conception. Importantly, we tested whether arousal mediates the relationship between the probability of conception and creative potential. As creativity is associated with originality and the aesthetic domain, we expected that divergent originality would be particularly enhanced when the probability of conception was high.

## 2. Materials and Methods

### 2.1. Participants

The study involved voluntary participation by 1045 Polish naturally cycling women aged 18 to 35 (*M* = 23.76; *SD* = 4.45) who were not using hormonal contraceptives. The sample size was assessed based on the recommendations of Gangestad et al. [94]. Due to his analysis, the best method based on a single session (an average of continuous forward and backward estimates) demands a sample size exceeding 700 participants. Gonzales and Ferrer [95] offer a similar recommendation. Women declared that they were not pregnant, had not given birth within the previous three months and were not breastfeeding a child. We excluded 294 participants who did not confirm these conditions. The final sample consisted of 751 participants. They were recruited via the Sona system at the SWPS University of Social Sciences and Humanities and via a social media webpage (www.facebook.com accessed on 17 May 2021). The research was approved by the Departmental Committee on Research Ethics. All data are freely available at: https://osf.io/8v7s9/ (accessed on 17 May 2021).

### 2.2. Materials

#### 2.2.1. Probability of Conception

To calculate the probability of conception for every participant, we applied an estimated distribution, which allowed us to calculate the day-specific probability that conception could occur [96]. The method provided estimates of day-specific probabilities of conception in fertile cycles with the use of ultrasound foetal biometry in the first trimester as a proxy in a large cohort of spontaneous singleton pregnancies [96] based on the assumption that the day of conception in pregnant women may be considered the day of ovulation. The method allows us to estimate the probability of conception in both regular and irregular cycling women, using also the information about their age. The regularity was a bicategorical variable created from the declaration (yes/no) and the cycle length designation. On the basis of the date of survey participation and the self-reported data about the first day of the last period, we calculated the specific day of the cycle for every participant. Due to the table of day-specific probabilities proposed by Stirnemann [80], we established the probability of conception rates, which we applied to statistical analyses. The data we used and the day-specific probabilities table are included in the study database freely available at: https://osf.io/8v7s9/ (accessed on 17 May 2021).

#### 2.2.2. Arousal

To test women’s arousal, we used the Self-Assessment Manikin (SAM), which is a picture-oriented questionnaire measuring an emotional response [97]. We used two of the three subscales: valence/pleasure was assessed (5 pictures ranging from the most negative = 1 to the most positive = 5), arousal was assessed (rated from low = 1 to high = 5) and dominance (the third subscale) was disregarded. The participants were asked to match the picture that most closely corresponded to their state.

#### 2.2.3. Divergent Thinking

Divergent thinking is a valid indicator of creative potential in various contexts [98]. We used the task by Corbalan and Lopez [99] as a proxy to measure divergent thinking performance. The authors linked creativity to the capacity to ask appropriate questions. The task we applied involved an ambiguous picture that may facilitate women’s mating imagery. The picture presents a social situation with a couple dancing in the foreground (the picture is included at https://osf.io/8v7s9/ (accessed on 17 May 2021). The participants were asked to generate as many questions about the picture content, while trying to be creative as they could be within a 5 min period [100]. The participants’ ideas were aggregated and scored based on fluency, flexibility and originality by four independent raters.

### 2.3. Procedure

Data were collected online using the Qualtrics research platform. The women were informed they were participating in a study on the relationship of mental associations during the ovulatory cycle. They were initially asked to mark the valence of their mood and the level of experienced arousal. After the ambiguous picture was presented, the women were asked to generate questions about what the picture was about in a 5 min period. The questions were saved in the dialog window behind the picture so that the women could see the picture for the whole 5 min. The data were coded by four raters for fluency (the sum of all nonredundant ideas presented), flexibility (the number of semantic categories applied) and originality (quantitative measure of uncommonness, which was the average originality rating of all creative answers). At the end, the participants were asked about the first day of their last menstruation, the average length of their ovulatory cycle, the regularity of their cycles and their history of pregnancy, childbirth and breastfeeding within the past three months. They were also asked to report their age.

## 3. Results

We first calculated descriptive statistics for all variables and examined Pearson’s correlations between these variables using IBM SPSS Statistics 25. We also calculated Kendall *W* for the interrater reliability of creativity assessments. In the next step, we conducted a mediation analysis using Model 4 PROCESS [101]. The indirect effects were tested with bias-corrected bootstrapping (*n* = 5000) and 95% confidence intervals (*CI*) for the indices. When a 95% bootstrapped *CI* does not include zero, it indicates that the parameter is statistically significant.

### 3.1. Initial Analyses

We first computed Kendall *W* for the interrater reliability of the divergent thinking scores. For fluency, the reliability was *W* = 0.99, *p* < 0.001; for flexibility, *W* = 0.75, *p* < 0.001; and for originality, *W* = 0.72, *p* < 0.001. These results indicated rater cohesion.

In the next step, we computed descriptive statistics and correlation coefficients between all continuous variables. We present the results in Table 1 and Figure 1. We found that the probability of conception was positively correlated with creative originality (*r* = 0.24, *p* < 0.001) and creative flexibility (*r* = 0.07, *p* = 0.036). This means that as the probability of conception increased, both creative originality and flexibility also increased. However, the probability of conception was not correlated with creative fluency (*p* = 0.166). All dimensions of divergent thinking were positively correlated: originality with fluency (*r* = 0.49, *p* < 0.001) and flexibility (*r* = 0.53, *p* < 0.001) and fluency with flexibility (*r* = 0.79, *p* < 0.001). Arousal and valence were correlated (*r* = 0.18, *p* < 0.001), but surprisingly, arousal did not correlate with any other variables (*p*_s_ > 0.248), however the association between arousal and the probability of conception was marginally significant (*r* = 0.05, *p* = 0.076). We found positive correlations between age and fluency (*r* = 0.08, *p* = 0.080), flexibility (*r* = 0.10, *p* = 0.006) and originality (*r* = 0.11, *p* = 0.002) and thus we conducted all analyses controlling for age of the participants.

### 3.2. Mediation Analysis: Arousal as a Mediator of the Relationship between Probability of Conception and Creative Originality

To determine whether arousal mediated the relationship between the probability of conception and creative originality, we conducted a mediation analysis using Model 4 PROCESS [101]. We introduced the probability of conception as a predictor, creative originality as the outcome variable and arousal as a mediator. As recommended by Hayes [101], the regression/path coefficients were unstandardized. As we found age positively correlated with all dimensions of divergent thinking, we included age in the analysis, as a control variable. Data from this analysis are presented in Table 2.

The analysis revealed that the direct effect between the probability of conception and originality was statistically significant, *b* = 0.069; 95% *CI* = [0.049, 0.090], and was maintained on the same level after we controlled for arousal, *b* = 0.069; 95% *CI* = [0.049, 0.090]. This indicated that the respondents with a higher probability of conception had more original ideas. However, when we controlled for the probability of conception, arousal did not influence originality, *b* = 0.008; 95% *CI* = [−0.080, 0.095]. The probability of conception did not predict the reported arousal levels, *b* = 0.012; 95% *CI* = [−0.005, 0.029]. The indirect effect was not significant, *b* = 0.00; 95% *CI* = [−0.001, 0.002]. Age significantly predicted originality, *b* = 0.029; 95% *CI* = [0.011, 0.047], with the same coefficient value both in partial and total model, but there was no relationship between age and arousal, *b* = 0.012; 95% *CI* = [−0.003, 0.026]. We can assume that, regardless of age, the self-reported arousal did not mediate the relationship between conception probability and creative originality of ideas.

### 3.3. Mediation Analysis: Arousal as a Mediator of the Relationship between Probability of Conception and Creative Flexibility

As the probability of conception correlated also with flexibility, we conducted a mediation analysis using Model 4 PROCESS [101]. We introduced the probability of conception as a predictor, the creative flexibility as the outcome variable and arousal as a mediator. We included age as a control variable. The results of these analysis are presented in Table 2.

The direct effect of the probability of conception was not significant, *b* = 0.035; 95% *CI* = [−0.004, 0.074], and did not change when we controlled for arousal, *b* = 0.036; 95% *CI* = [−0.003; 0.075]. The indirect pathway was not significant, *b* = 0.001, 95% *CI* = [−0.001, 0.005]. Age positively predicted flexibility in the partial model, *b* = 0.049; 95% *CI* = [0.015, 0.082], and changed to *b* = 0.049; 95% *CI* = [0.016, 0.083] in the total model. There was no relationship between age and arousal, *b* = 0.012; 95% CI = [−0.003, 0.026]. We can assume that, regardless of age, the self-reported arousal did not mediate the relationship between conception probability and creative flexibility of thinking.

## 4. Discussion

The study aimed to test the creative potential of women across the menstrual cycle. Specifically, we tested whether creative fluency, flexibility and originality increased with the probability of conception during the ovulation cycle. Importantly, we investigated the mediating role of arousal in the relationship between the probability of conception and creative thinking. This expectation was in line with Miller’s [44] signaling theory, in which creativity evolved through sexual selection as an indicator of one’s fitness and willingness to mate and was used in courtship to attract a potential mate. Many studies have shown that the phase of ovulation is the time when women indeed signal their fertility status, in an attempt to attract the attention of men [see 86 for review]. Therefore, if creativity serves as this type of signal, it should increase during periods of heightened sexual motivation that occur during the fertility peak [87].

Indeed, previous findings have suggested that women’s creative potential is enhanced during the ovulation phase [12,42,43]. Our study replicated these findings in more rigorous methodological conditions, as according to sample size recommendations [94], Krug’s [12,42,43] studies were underpowered (*N* ≈ 17). We applied a divergent thinking paradigm, which defines creative potential as fluent and flexible thinking leading to original ideas. This is one of the most commonly used paradigms in the psychology of creativity [102] and reflects the creative potential that can be found in everyday life and not necessarily among creative geniuses [78]. Specifically, we found that the probability of conception positively correlated with the creative originality of ideas and flexibility of thinking but not with the fluency of thinking. This indicated that the higher the probability of conception was, the more original ideas were generated. These ideas were also more varied and included more frequent changes in perspective, although they were not more numerous. However, although the probability of producing original ideas increases with the total number of generated ideas, the mere number of ideas is not essential for creativity. A creative person may produce only one idea, but it may be an original one [79].

The second goal of our study was to test the mediating role of arousal in the relationship between the probability of conception and creative thinking. Arousal seemed to be a good candidate for explaining the relationship between the ovulatory phases and creativity because (1) women are more aroused during ovulation than in the non-fertile phases of the ovulatory cycle [12,103], partially as a result of dopamine outputs correlating with estradiol [104], and (2) arousal enhances creativity according to a meta-analysis, see [105]. Although it has been previously suggested that arousal can mediate this effect [12], no study has directly tested this possibility. However, we found no association of arousal with the probability of conception on any dimension of creativity. This may suggest that self-reported measures of arousal are too overt to explain the evolutionarily driven mechanism. Additionally, women do not need to be aware of their arousal for it to affect their cognition or behavior. Therefore, we suggest that self-reported measures of arousal should be replaced with physiological measures to better track potential arousal-based mechanisms of the observed effects.

The results of our study are consistent with Miller’s hypothesis [44] in that enhanced creative potential during the ovulation phase is associated with reproduction and the mating goals, which are naturally activated during the fertile window when the possibility of conception is the highest. Specifically, creative originality can be associated with the nature of signaling, as this aspect corresponds to uncommonness and novelty, which could stimulate the senses and attract the attention of the beholder. Our results also resonate with the studies conducted by Griskevicious et al. [58], which showed that women preferred to signal their creativity to attract a prospective mate, at least in long-term mating conditions. Further studies are needed to test if creativity can be used as an advantage in intrasexual competition. Divergent thinking leads to many potential solutions, so it may enhance a competitor’s chance to achieve the goal of mating. There is also a relationship between estradiol and implicit power motivation and dominance [106]. This can be associated with a dark side of creativity, as original thinkers can be more dishonest when pursuing their goal [107]. Evolutionary models predict that female intrasexual competition occurs when a fit and healthy man is considered a resource [108], so testing this assumption in the context of finding a potential partner should be involved. Furthermore, creative originality may be a tactic to attract mates and compete with rivals similarly to make-up usage [109].

In the light of these study findings, some environmental factors associated with female fertility should be considered, as they can potentially influence cognitive abilities and, as a result, impair women’s reproductive systems. Of the utmost importance are those that affect human oocyte quality and function, and contribute to accelerated oocyte aging and female infertility [5]. These include a high-fat diet, as obesity can lead to anovulation, polycystic ovarian syndrome, or pregnancy complications [110]. Poor diet can also lead to insulin resistance and the resulting hyperinsulinemia, which can facilitate ovulatory dysfunction and infertility [5,111]. Environmental pollution factors are of no less importance. Women’s exposure to pesticides on a daily basis is considered a significant risk factor in fertility problems [5]. Pesticides, such as Mancozeb, can lead to delayed ovulation and menstrual cycle disruptions, and as such impede female fertility (see [112], for a review). These results lead to a worrying conclusion. Toxic agents that disrupt our natural environment may also lead to a degradation of mechanisms shaped in humans by evolution, including those related to fertility.

The other important issue relates to combined oral contraceptives, which are commonly prescribed to women of reproductive age. Data on the impact of contraceptives on cognitive abilities are contradictory [113]. Neuroscientific research indicates socio-emotional effects of contraceptive usage, as they can impact emotional reactivity and fear-learning systems, as well as social functions, such as partner preference [114]. Studies point to increased rates of depression, anxiety, fatigue, neurotic symptoms, compulsion and anger [115]. These emotional effects are supposed to be related to the hormone type dominating in the pill, as estrogens may enhance mood, while progestins may decrease it [114]. These effects may potentially affect creativity via emotional and motivational states, as the influence of these states is vastly explored in the psychology of creativity [116].

## 5. Limitations of the Study

The limitations of the study must be outlined. First, we applied self-reported cycle data with no verification of the dates of menstruation. Although participants were instructed to recall the date of their last menstruation as a study preparation to augment the accuracy of declarations, it is still not a reliable method. However, our sample size meets the recommendations of Gangestad et al. [94] for conducting measurements using self-reported cycle data and sufficient power of a between-subject study. Future studies should apply the within-subject design rather than between-subject with the use of hormonal measurements assessing the ovulatory cycle phase [94].

Secondly, the assessment of women’s arousal has limitations as we used a declarative measure. Although the Self-Assessment Manikin has proven its reliability [97], there is a possibility that slight changes in the level of arousal may not be consciously detectable by participants. Thus, it is essential to turn to physiological measurements of arousal in the further investigations. It should be noted that fertility-related shifts in mating psychology may depend on relationship status and within-relationship factors such as partner’s attractiveness or relationship satisfaction [117,118,119]. We did not collect such data, as the procedure of our study did not involve any overt context of romantic relationships. There could also have been other factors potentially covarying with women’s creativity or fertility that we did not control. For example, general level of stress may impair both creative thinking [120] and women’s fertility [121]. We did not control for psychiatric and neurological diseases, nor for taking neuroactive medicine. Additionally, as experiencing pain may affect cognitive abilities [122], it would be beneficial in future studies to consider the painful symptoms that may accompany menstruation. Furthermore, as creativity correlates with intelligence, the level of education can play a role in between-subject comparisons. However, in our study, the participants were recruited mostly from an academic environment, so the education factor was not differentiated. The limitation of the study is also its Internet-based procedure, which naturally prevented standardization of situational conditions in which the participants completed the survey.

## 6. Conclusions

Creative thinking is a defining human feature. It provides novel solutions and allows us to look at our life from different perspectives. As such, it can attract attention and serves as a potential fitness indicator. If this is indeed the case, we should find a relationship between the probability of conception and women’s creative potential. The results of our study indicate that the higher the probability of conception during the ovulation cycle, the more original and flexible the thinking of the women was. However, we did not confirm a mediating role of arousal in this relationship. Future studies should adopt more sophisticated measures of both arousal and the probability of conception.

## Figures and Tables

**Figure 1 ijerph-18-05390-f001:**
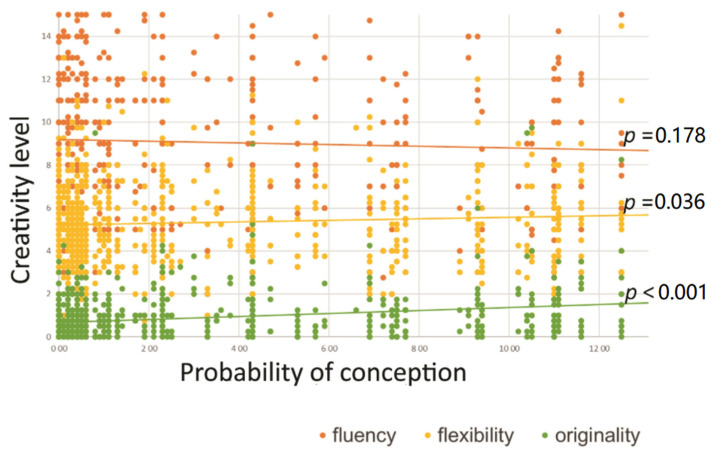
The linear relationship between the probability of conception and creative fluency, flexibility and originality.

**Table 1 ijerph-18-05390-t001:** Means, standard deviations and correlations among study variables, controlling for age, *N* = 751.

Variables	*M*	*SD*	1	2	3	4	5
1. Creative fluency	9.06	4.67					
2. Creative flexibility	5.32	2.1	0.79 ***				
3. Creative originality	0.9	1.13	0.49 ***	0.53 ***			
4. Arousal	2.67	0.9	−0.00	0.04	0.02		
5. Valence	3.5	0.87	−0.00	−0.01	0.01	0.18 ***	
6. Probability of conception	3.15	3.85	−0.04	0.07 *	0.24 ***	0.05	0.00

Note: Cell entries are zero-order correlation coefficients one-tailed, * *p* < 0.05, *** *p* < 0.001.

**Table 2 ijerph-18-05390-t002:** Direct and indirect effects coefficients of the mediation analyses of the relationship between the probability of conception and creative originality and flexibility, with arousal as a mediator and age as a controlled variable.

Pathways	Originality						Flexibility					
	*b*	*SE*	*t*	*p*	*CI* 95%		*b*	*SE*	*t*	*p*	*CI 95%*	
					*LL*	*UL*					*LL*	*UL*
Direct												
a	0.012	0.009	1.414	0.158	−0.005	0.029	0.012	0.009	1.414	0.158	−0.005	0.029
b	0.017	0.045	0.383	0.702	−0.071	0.105	0.099	0.085	1.175	0.240	−0.067	0.267
c’	0.069	0.011	6.545	<0.001	0.048	0.089	0.035	0.020	1.748	0.081	−0.004	0.074
c	0.069	0.011	6.578	<0.001	0.048	0.090	0.036	0.020	1.811	0.071	−0.003	0.075
Indirect												
	0.000	0.001			−0.001	0.002	0.001	0.002			−0.001	0.005

Note: *b* = unstandardized mediation coefficient; *SE* = standard error; *t* = t-test coefficient; *p* = level of significance (two-tailed); *CI 95%* = confidence interval presented as bias-corrected and accelerated 5000 bootstrapping; *LL* = lower limit; *UL* = upper limit.

## Data Availability

Data are freely available at: https://osf.io/8v7s9/ (accessed on 17 May 2021).

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
