# Peer review of "The More Fertile, the More Creative: Changes in Women’s Creative Potential across the Ovulatory Cycle"

_ijerph, 2021, doi:10.3390/ijerph18105390_

Round 1

Reviewer 1 Report

This study explores the relationship between the probability of conception during the ovulation cycle and creative potential. The paper has a clear motivation and presents interesting results.  There are some of the issues that require attention.

  1. The literature review throughout the paper presents a suspiciously clear cut and consistent body of scientific literature. I suspect this is a misrepresentation. For instance, contrary to the literature review suggesting that women at peak fertility show greater interest in uncommitted sexual relationships, a newer study has failed to detect this effect (https://doi.org/10.1177/1474704920976318). In general, the claims that “fertility status affects women’s mating psychology” are exaggerated. There is a growing number of papers showing a lack of cyclical shifts in women’s mating psychology and behavior.
  2. Maybe I missed something, but is there any evidence that women’s creativity attracts men?
  3. The probability of conception calculation needs to be made clear enough that another researcher could replicate them. How cycle regularity was defined?
  4. The table 2 is quite hard to follow. The reader would benefit from a more throughout explanation in the table legend.
  5. In Figure 1, description: “The linear relationship between the probability of conception and creative fluency, flexibility and originality potential solutions to problems” suggests that all correlation are significant. Could you please add p value for each line?
  6. The literature on creativity across the adult life span indicates a decline in creative activity with increasing age. Therefore age should be included in models as a confounding factor. What about other factors that may affect creativity? If these data are not available, they need to be thoroughly discussed as limitations of the study.
  7. Limitations of the paper should be clearly outlined. Especially validity of assessments of conception probability in ovulatory cycle and reliance on counting methods for cycle phase position should be discussed.
  8. Finally, the authors suggest that creativity may be useful as a signal for potential partners (intersexual selection mechanism). Can creativity give women an advantage in intrasexual competition (intrasexual selection mechanism)? It's worth considering in the introduction and discussion.

Reviewer 2 Report

Reviewer comments on Galasinska et al.:

Article: The more fertile, the more creative: Changes in women’s creative potential across the ovulatory cycle.

The aim of this manuscript is to test the creative potential of women during the menstrual cycle, discussing the results in terms of signalling theory and from an evolutive point of view. The second goal of this study is to test the mediating role of arousal, in the relationship between the probability of conception and creative thinking.

Please find below an enumerated list of comments on my review of the manuscript:

INTRODUCTION:

LINE 23:

Reasonable, it is important to add to this introductive section that the optimal fertilization window for women appears about 12-14 hours after ovulation, as reported in this study (Ultrastructural markers of quality are impaired in human metaphase II aged oocytes: a comparison between reproductive and in vitro aging -2015).

LINE 25:

The manuscript may benefit from highlighting how the ovulatory cycle’s characteristics (e.g. lenght, hormonal balance) depend from a sophisticated cross-talk between ultrastructural (https://link.springer.com/chapter/10.1007/5584_2019_456#:~:text=With%20aging%2C%20there%20is%20a,ovarian%20reserve%20and%20oocyte%20quality), cellular, molecular and environmental factors (Association between female reproductive health and mancozeb: Systematic review of experimental models – 2020).

MATERIAL AND METHODS:

As regards this section, the methodology design was appropriately implemented within the study.

RESULTS:

This section is well organized and densely presented, based on well-synthetized data.

DISCUSSION:

LINE 179-181: 

Consider also including the role and the impact of the environmental pollution on the ovulatory shift and, consequently, on female creativity. Several studies highlight the association between female reproductive health and the exposure to environmental pollutants, with severe and deleterious effects on the female reproductive competence (https://pubmed.ncbi.nlm.nih.gov/27887783/; Association between female reproductive health and mancozeb: Systematic review of experimental models – 2020).

LINE 190: shownthat oestradiol for SHOWN THAT OESTRADIOL.

To sum up, the topic is timely and call for attention. Overall, the manuscript requires minor changes (as mentioned). I would accept the manuscript, if the comments are addressed properly.

Round 2

Reviewer 1 Report

Thank you for the changes made. All replies to my remarks are sufficient.